# Effects of Hypopressive Abdominal Training on Ventilatory Capacity and Quality of Life: A Randomized Controlled Trial

**DOI:** 10.3390/healthcare12090893

**Published:** 2024-04-25

**Authors:** Maria del Carmen Herena-Funes, Caroline Correia de Alencar, Dara María Velázquez-Torres, Elisenda Marrero García, Yolanda Castellote-Caballero, Felipe León-Morillas, Aday Infante-Guedes, David Cruz-Díaz

**Affiliations:** 1Department of Health Sciences, Faculty of Health Sciences, University of Jaén, 23071 Jaén, Spain; mchf0001@red.ujaen.es (M.d.C.H.-F.); dcruz@ujaen.es (D.C.-D.); 2Grupo ICOT Arnao (Gran Canaria), 35200 Telde, Spain; 3Grupo ICOT Vecindario (Gran Canaria), 35110 Vecindario, Spain; 4Faculty of Health Sciences, University of Atlántico Medio, 35017 Las Palmas de Gran Canaria, Spain; aday.infante@pdi.atlanticomedio.es; 5Department of Physiotherapy, Faculty of Physiotherapy, Podiatry and Therapy Occupational, Catholic University of Murcia (UCAM), Guadalupe, 30107 Murcia, Spain; fleon@ucam.edu

**Keywords:** hypopressive abdominal exercises, ventilatory capacity, quality of life

## Abstract

Pelvic floor dysfunctions, associated with alterations in respiratory mechanics and, consequently, quality of life, are the cause of the most frequent gynecological problems. Pelvic floor muscle training emerges as a first-line treatment, with new approaches such as hypopressive exercises. The aim of this study was to analyze the efficacy of an 8-week supervised training program of hypopressive exercises on the pelvic floor and its impact on improving the ventilatory mechanics and quality of life in women. Analysis of the spirometric parameters showed a significant main Group × Time effect for three parameters: the ratio of FEV_1_/FVC (*p* = 0.030), the forced expiratory flow at 75% of the expired vital capacity (*p* < 0.001), and the forced expiratory flow over the middle half of the forced vital capacity (*p* = 0.005). No statistical significance was found regarding the SF-12 questionnaire components; only differences were found over time in the physical role (*p* = 0.023), bodily pain (*p* = 0.001), and vitality (*p* < 0.010) domains and in the physical component summary score (*p* = 0.010). After an 8-week intervention of hypopressive exercises, an improvement in the ventilatory and pulmonary capacities can be observed.

## 1. Introduction

Understanding health as a state of complete psychological, physical, and social well-being, and not merely as the absence of disease or infirmity, contributes a positive notion to the concept of quality of life (QoL) [1,2]. Pelvic floor dysfunctions (PFDs) are associated with a decrease in QoL, with social, healthcare, and economic repercussions [3,4]. However, they are often given low priority compared to other health problems simply because they are not considered life-threatening [5]. It is estimated that 25% of women experience PFDs, a figure that increases with age to 53% [6]. Between 3% and 6% of women are estimated to develop some of these symptoms over their lifetimes. According to World Health Organization (WHO) forecasts, by the year 2050, about one-third of women aged between 45 and 65 years will suffer from PFDs [7].

The etiology of PFDs is multifactorial and strongly associated with obstetric factors and aging [4,8]. An important predisposing factor is the increase in intra-abdominal pressure (IAP), which can be significantly influenced by anthropometric factors such as obesity, and chronic conditions, including constipation and chronic cough [4]. Additionally, the muscular function plays a critical role, requiring a proper balance in the activation and functioning of the abdominal muscles, pelvic floor, and diaphragm. This balanced interaction is essential for maintaining optimal IAP levels [1,4]. The diaphragm and pelvic floor muscles work in synergy: when the diaphragm contracts, it descends along with the pelvic floor in an eccentric contraction that increases the IAP [9,10]. This dynamic illustrates the intricate relationship between the state of the pelvic floor musculature and a patient’s respiratory mechanics. However, despite the close relationship between intra-abdominal pressure (IAP) and some pelvic floor dysfunctions (PFDs), it is important to note that a cause–effect relationship between IAP and PFDs cannot be established universally. It is crucial to acknowledge that some dysfunctions, such as dyspareunia, are not as heavily influenced by the management of IAP [11]. Nonetheless, for proper pelvic floor health and functional activity, a comprehensive approach is essential that includes a thorough evaluation of the Lumbopelvic Abdominal Complex (LPAC). This assessment should encompass the condition of the diaphragm, pelvic floor, and abdominal wall to ensure an integral treatment plan if the patient exhibits this type of condition [4].

The relationship between the ventilatory capacity and pelvic floor health is a critical area of study within women’s health, elucidating the interconnectedness of the respiratory and pelvic floor functions. The diaphragm and pelvic floor muscles function synergistically: as the diaphragm contracts and descends during inspiration, the pelvic floor muscles also descend, managing the intra-abdominal pressure effectively. This coordinated movement is essential for both respiratory efficiency and pelvic stability [12]. Research has demonstrated that alterations in respiratory mechanics, such as those caused by chronic respiratory conditions or improper breathing patterns, can lead to increased intra-abdominal pressure, subsequently contributing to pelvic floor dysfunctions, including incontinence and prolapse [13]. Moreover, empirical evidence supports the proposition that targeted interventions in pelvic floor and respiratory muscle training not only enhance the strength and functionality of pelvic floor muscles but also improve ventilatory capabilities. This suggests a reciprocal benefit, wherein strategies designed to augment one function are likely to positively impact the other. Such findings advocate for an integrated approach in clinical practices that aims to optimize both respiratory and pelvic floor health [14]. This bidirectional relationship underscores the importance of an integrated approach in the assessment and treatment of pelvic floor disorders, promoting the inclusion of respiratory assessments in pelvic health evaluations.

Currently, there are various interventions for strengthening the pelvic floor, among which hypopressive techniques stand out. These techniques are based on movement control through breathing [15]. The sequence of hypopressive techniques usually lasts between 20 and 60 min using different positions. In each position, three diaphragmatic breaths and one hypopressive breath are performed, based on blocking the diaphragm muscle (apnea) and contracting the abdominal muscles [16,17]. Although the results obtained through hypopressive training have not demonstrated superiority over traditional active approaches to pelvic floor strengthening, the evidence suggests that hypopressive training can achieve improvements in the endurance and muscular strength of the pelvic floor muscles as well as the deep-trunk musculature [16,17]. Beyond the direct activation that occurs in the pelvic floor during hypopressive exercises, there is theoretical discussion about a possible synergistic influence between the activity of the transverse abdominal muscle and the pelvic floor musculature [17].

To the best of our knowledge, there is limited published evidence on the analysis of the effects of respiratory muscle training on the lung function and quality of life in women with pelvic floor dysfunction. Therefore, our objectives were to investigate the influence of 8 weeks of hypopressive abdominal training on the ventilatory and pulmonary capacities in women with PFDs, as well as to study the effects of this intervention on the health-related quality of life in women post-intervention.

## 2. Materials and Methods

### 2.1. Study Design

This study was a blinded randomized controlled trial (RCT), officially registered on Clinicaltrial.gov (NCT04343599). It was conducted from February to June 2019 at the University of Jaén, Spain, and adhered to the Consolidated Standards of Reporting Trials (CONSORT) [18] statement guidelines and the Consensus on Exercise Reporting Template (CERT) [19]. All subjects agreed to take part in the study and signed the corresponding informed consent.

### 2.2. Participants

A randomized controlled trial was performed. Participants were recruited by advertisements in public institutions (hospitals and universities) and on social media (Instagram and Facebook). Eligible participants were women aged between 18 and 60 years who had been experiencing pelvic floor dysfunction (PFD) symptoms, assessed using the Pelvic Floor Distress Inventory (PFDI-20) and the Pelvic Floor Impact Questionnaire (PFIQ-7) [20], for more than 6 months. Exclusion criteria included any prior engagement with Home Exercise (HE), receiving conservative treatment for PFD in the previous year, undergoing pelvic or abdominal surgery, or having medical conditions that contraindicate participation in HE, such as pregnancy, uncontrolled hypertension, hiatal hernia, or cardiorespiratory disease.

### 2.3. Randomization and Masking

Following the established criteria, participants were randomly assigned to either the experimental group (EG) or control group (CG) using the Oxford Minimization and Randomization (OxMaR) system at a 1:1 ratio [21]. Randomization was conducted by an independent researcher who was not involved in the data collection or intervention. Regarding the data collection, the researcher was blinded to the group assignments of the participants.

### 2.4. Intervention

Participants in the intervention group participated in a Hypopressive Therapy program, adhering to the Low-Pressure Fitness (LPF) protocol. This regimen consisted of two weekly sessions, each lasting 30 min, for a total duration of eight weeks [22]. The initial two sessions were focused on imparting the essential breathing techniques and postural foundations of hypopressive exercises, as elaborated in Table 1.

These core principles were consistently applied in all the subsequent postures and their variations, particularly involving the upper limbs, as illustrated in Figure 1 and Figure 2. During the intervention, participants engaged in the intervention group began with a warm-up targeting the respiratory, postural, and abdominopelvic musculatures. Subsequently, the main part of the training was conducted based on the postures outlined by the Low-Pressure Fitness (LPF) method, involving static postures and dynamic transitions during expiratory apnea. Moreover, while consistently maintaining the postural principles of the method, position changes from standing to sitting and quadruped were executed. The duration of expiratory apnea varied between 20 and 30 s depending on the participant’s endurance and the difficulty of the exercise. Each exercise was repeated three times during the 30 min session and, to ensure proper form, participants were instructed to maintain an intensity of 5–7 on the perceived exertion scale. The distribution of the sample into small groups (10 participants) facilitated personalized adaptations for each participant to determine the optimal progression strategy. Participants were excluded from the study if they missed more than two training sessions.

### 2.5. Methods

Forced spirometry was conducted using a commercial spirometer, the SP10 Contec, adhering to the protocol established by the Spanish Society of Pulmonology and Thoracic Surgery (SEPAR) [23]. This technique involves performing a maximum inspiration followed by a forced expiration. The variables analyzed were the forced vital capacity (FVC), the maximum volume expired in the first second of forced expiration (FEV_1_), the peak expiratory flow rate (PEF), the ratio of FEV_1_/FVC (FEF1%), the forced expiratory flow at 25% of the expired vital capacity (FEF25%), the forced expiratory flow at 75% of the expired vital capacity (FEF75%), and the forced expiratory flow over the middle half of the forced vital capacity in percentage (FEF25–75%) [24,25,26].

Health-related quality of life was assessed through the Spanish version of the Short Form-12 Health Survey (SF-12), obtained through a multiple regression of the SF-36 proposed by Grandek et al. [27]. This questionnaire consists of 12 items that encompass 8 dimensions (2 items per dimension): physical functioning, physical role, bodily pain, general health, vitality, social functioning, emotional role, and mental health. These dimensions are coded and transformed into a scale ranging from 0 (the worst health state for that dimension) to 100 (the best health state). This questionnaire was adapted to the Spanish version and validated by Vilagut et al. [28], and it has shown a reliability score of 0.70.

### 2.6. Data Analysis

In this study, statistical analysis was performed using the SPSS software, version 21.0. Categorical variables are described with frequencies and percentages, while continuous variables are defined with means and standard deviations. The Kolmogorov–Smirnov test was used to verify the normality of the continuous variables. For the descriptive analysis of the quantitative (continuous) and qualitative (categorical) variables, the Student’s *t*-test and chi-square test were used, respectively. To segregate the variability in the results, a 2 × 2 Analysis of Variance (ANOVA) was employed, wherein the between-group component was determined by the hypopressive abdominal training (experimental group vs. control group) and the within-group component was the time of measurement (pre- and post-intervention). Eta-squared was used for the effect size of the main effects, and Cohen’s d statistic was applied to calculate the effect size of the specific Group × Time interactions. An effect size difference was considered: negligible: <0.2; small: between ≥0.2 and ≤0.5; moderate: between ≥0.5 and ≤0.8; and large: ≥0.8 [29].

As dependent variables, the ventilatory capacity (spirometric parameters) and health-related quality of life (SF-12) were taken. An independent data analysis was conducted for each dependent variable. For the dependent variable “ventilatory capacity”, pre-test differences were detected in three values (FVC, FEV1%, FEF75), leading to an Analysis of Covariance (ANCOVA) taking the pre-test value as a covariate and the statistical analysis of the post-test inter-group differences (only carried out for FEF75 because in the other two values, the effect of the Group × Time interaction was not significant). Results were considered statistically significant with a *p*-value < 0.05. The sample size was calculated using Ene 3.0 (GlaxoSmithKline, SA, Madrid, Spain) [30], based on previous studies [17,31], to obtain a statistically significant difference using stabilometric scores as the dependent variable, with a power of 0.80, a significance level of 95%, and considering an estimated dropout of 15%. A total of 53 participants per group were required.

## 3. Results

### 3.1. Selection Process

A total of 254 participants were selected, of whom 76 did not meet the inclusion criteria and 53 did not agree to participate in the study. A total of 125 women were randomly assigned to each group (64 to the EG and 60 to the CG); five CG participants did not complete the last assessment and two EG women dropped out due to time incompatibility. Finally, a total of 117 participants completed the study with a mean age of ±45.65 years (Figure 3 and Table 2).

### 3.2. Respiratory Muscle Function

The respiratory function results after conducting a 2 × 2 ANOVA showed a significant main effect of Time for all the variables. Regarding the Group factor, there were significant changes in the FEF75, FEF1%, and FEF25–75% (Figure 4); the rest of the variables for this factor did not show significant changes.

A significant main Group × Time effect was observed for three parameters: the FEV1%, FEF75, and FEF25–75. A subsequent detailed analysis of the Group × Time interactions for these three variables identified significant pre- and post-measurement differences for the experimental group in the following variables: the FEV1%: t(62) = −2.216, *p* = 0.030, Cohen’s d = 0.10; FEF25–75: t(62) = −3.447, *p* = 0.001, Cohen’s d = 0.30; and FEF75: t(62) = −3.076, *p* = 0.003, Cohen’s d = 0.39. Similarly, this specific analysis showed that there were significant differences between both groups in the post-intervention measurement, both for the variable FEV1% (t(117) = −2.483; *p* = 0.014; Cohen’s d = 0.32) and the variable FEF25–75 (t(117) = −2.841; *p* = 0.005; Cohen’s d = 0.37). Because there were significant differences between the groups regarding the FEF75 variable in the pre-intervention analysis, an ANCOVA was conducted for the analysis of the post-intervention differences, which observed that the experimental group was significantly superior to the control (*p* < 0.001, η^2^ = 0.291) (Table 3). Moreover, the obtained data suggest that the study participants can be considered within normal limits, achieving FEV_1_ values ≥ 80% of the predicted value after adjusting for a person’s age, gender, and ethnic background.

### 3.3. Quality of Life (QoL)

Regarding the health-related QoL, the study results from the SF-12 questionnaire components showed significant changes due to the main Time effect in the physical role domain: F(1,117) = 5.27, *p* = 0.023, η^2^ = 0.043; bodily pain domain: F(1,117) = 12.79, *p* = 0.001, η^2^ = 0.099; and vitality domain: F(1,117) = 13.09, *p* < 0.001, η^2^ = 0.101, as well as in the MSC summary score: F(1,117) = 6.88, *p* = 0.010, η^2^ = 0.056. The rest of the effects showed no significant changes (Table 4).

## 4. Discussion

Our study aimed to analyze the influence of an 8-week regimen of hypopressive abdominal training on the ventilatory and pulmonary capacities in women with pelvic floor dysfunctions (PFDs), as well as to evaluate the effects of this intervention on the health-related quality of life post-intervention. Our findings suggest that engaging in a hypopressive abdominal exercise protocol offers more significant benefits for the respiratory function than not undergoing any treatment at all. These findings align with the established research, highlighting the significant impact of hypopressive abdominal exercise on improving the ventilatory function. This association opens up valuable therapeutic possibilities for enhancing respiratory health, presenting critical clinical implications for those dealing with respiratory disorders [32].

Our analysis revealed statistically significant enhancements in the FEF1% and FEF25–75% indices following the intervention, marking these improvements as clinically relevant. The FEF1% index is extensively employed and validated for the assessment of bronchoconstriction in both the large and intermediate airways, whereas the FEF25–75% index is indicative of the small airways’ condition. To the best of our knowledge, there is scant specific scientific evidence on the application of hypopressive exercises in adult populations or individuals with obstructive respiratory conditions. Nonetheless, the results observed in our intervention suggest that hypopressive exercises might benefit those with obstructive respiratory conditions by enhancing their lung capacities and breathing efficiencies. Additional evidence from studies on inspiratory muscle training, which shares physiological targets with hypopressive exercises, supports the potential for respiratory muscle strengthening to improve the overall pulmonary function [33]. This enhancement in the pulmonary function may alleviate symptoms associated with respiratory diseases, such as COPD and asthma, indicating possible benefits from hypopressive exercises [24].

Incorporating hypopressive respiratory maneuvers into treatments for PFDs may result in more favorable outcomes in terms of respiratory function parameters. Furthermore, considering the diaphragm’s role in postural stabilization, hypopressive abdominal exercise might offer improvements in both diaphragmatic and pelvic floor muscle functionality and their synergistic operation [34]. While most published evidence on the effects of hypopressive abdominal exercise concentrates on the pelvic floor musculature, addressing issues like incontinence, prolapse, or various pelvic floor dysfunctions [35,36,37], research on its impacts on the respiratory function remains scarce [38]. Our findings are in line with prior research that demonstrates that respiratory exercises, potentially including hypopressive techniques, can enhance exercise tolerance and reduce breathlessness in diseases affecting the airways, such as chronic obstructive pulmonary disease (COPD) [39]. Moreover, the improvements in the FEV1% and FEF25–75% indices could correspond to the enhancements observed in the ventilatory functions [40]. These correlations indicate that interventions targeting respiratory mechanics, such as hypopressive exercises, might have a significant and beneficial impact on various aspects of respiratory health, improving both the large- and small-airway functions.

Given these preliminary indications, additional research into the specific applications of hypopressive exercises for adults and patients with respiratory problems is warranted to establish definitive evidence and practical guidelines.

Regarding quality of life (QoL), we observed benefits in the physical role, bodily pain, and vitality, though these were not statistically significant in the experimental group. The results of this study did not differ significantly between the two groups. This outcome could be attributed to the fact that the baseline values for this questionnaire were already high, leaving limited room for improvement. Half of the participants in the experimental group (51.8%) did not engage in sports, suggesting that structured and supervised training itself is a key factor in improvement [31].

Our results suggest that hypopressive abdominal exercise protocols can be considered an intervention aimed at maintaining or improving the general psychophysical condition, the proper functioning of an overloaded body, or full recovery after illnesses, injuries, or states of respiratory fatigue, in line with previous research regarding its benefits [41]. Hypopressive abdominal exercise could maintain the physical condition and general health status of women. To the best of our knowledge, there has been no scientific evidence analyzing the effects of hypopressive abdominal exercise on the lung function and quality of life, making our findings potentially novel and not directly comparable to other studies.

An enhanced ventilatory capacity is increasingly recognized as a crucial factor influencing quality of life, especially in the adult population. Studies have shown that improvements in the ventilatory function significantly elevate physical independence and autonomy among adults, contributing to a more active and self-sufficient lifestyle [42]. Additionally, the inclusion of hypopressive training, which reduces intra-abdominal pressure and potentially strengthens respiratory muscles, may further enhance the quality of life and functional independence in this group [43]. Despite these promising findings, ongoing research is essential to fully delineate the benefits and optimize training protocols. Future studies must continue to explore these aspects to establish evidence-based practices that effectively integrate hypopressive exercises into health regimens [44].

## 5. Conclusions

Our findings indicate statistically significant improvements in the ratio of FEV_1_/FVC (FEV1%), the forced expiratory flow at 75% of the exhaled vital capacity (FEF75%), and the forced expiratory flow over the middle half of the forced vital capacity (FEF25–75%). These improvements highlight the potential benefits of hypopressive exercises in enhancing the ventilatory function in patients with obstructive airway disease. The clinical relevance of these parameters, particularly the FEV1%, which assesses bronchoconstriction in both the large and intermediate airways, and the FEF25–75%, indicative of small-airway health, suggests that hypopressive exercises may significantly impact respiratory mechanics. Given the promising outcomes observed in this study, further investigation into the broader applications of hypopressive exercises across different patient populations is warranted. Future research should aim to confirm these findings in larger, more diverse cohorts and explore the mechanisms by which these exercises affect respiratory health, ultimately guiding clinical practices and patient care protocols.

## 6. Strengths and Limitations

This study’s significant strength lies in its examination of the impact of hypopressive exercises on the lung function, contributing valuable data to a field wherein such interventions are less documented yet highly relevant. Our results help to elucidate the physiological changes that these exercises may induce in respiratory parameters, emphasizing their potential in managing obstructive airway conditions. Regarding limitations, while our study did not directly address the changes in pelvic floor dysfunction (PFD) symptoms, this was a deliberate scope limitation rather than an oversight. This research is part of a broader project that has already explored and published findings on pelvic floor variables [31]. The comprehensive approach of this larger project is designed to offer robust scientific evidence on a practice that is becoming increasingly widespread and shows promising results. By focusing on different aspects of hypopressive exercise across various studies within the project, we aim to provide a holistic understanding of its benefits.

## Figures and Tables

**Figure 1 healthcare-12-00893-f001:**
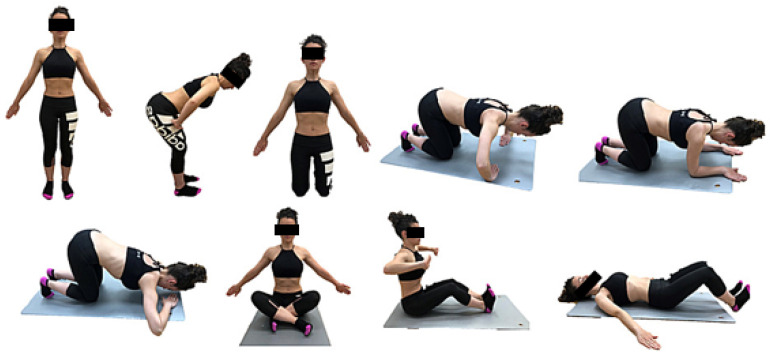
Progression of hypopressive postures.

**Figure 2 healthcare-12-00893-f002:**
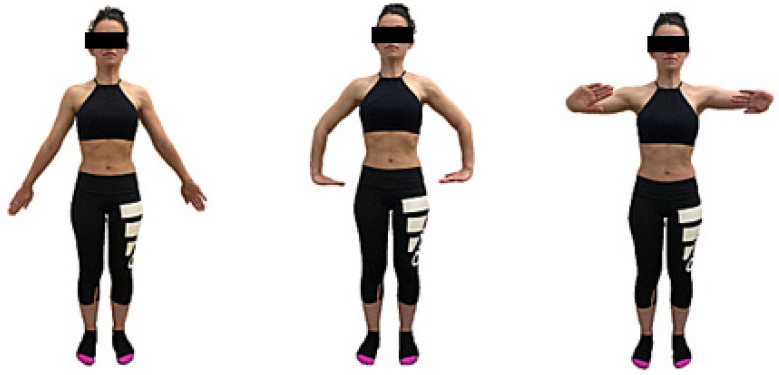
MMSS variant in hypopressive positions: (1) extension and pronation; (2) internal rotation shoulder, 90° flexion elbow, and wrist, hand push to the floor; (3) internal rotation and 90° flexion shoulder, 90° flexion elbow, and wrist push forward.

**Figure 3 healthcare-12-00893-f003:**
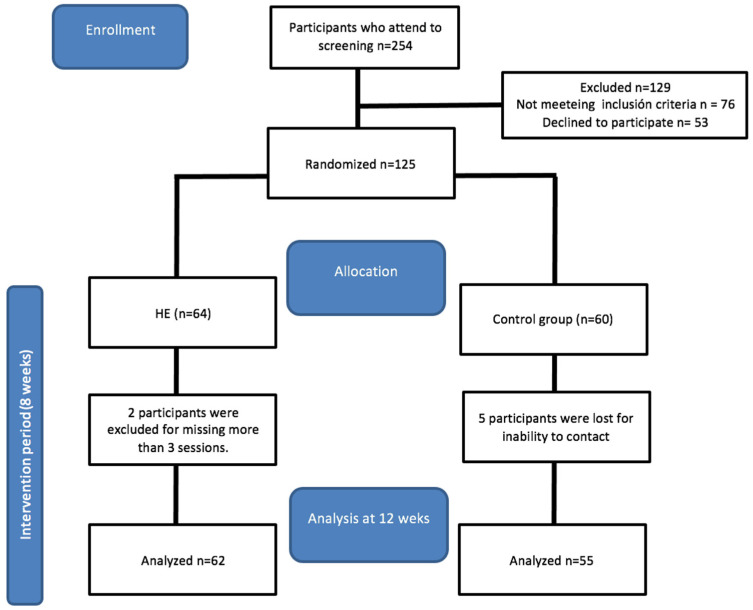
Flow chart of the study design and participant follow-up through the trial.

**Figure 4 healthcare-12-00893-f004:**
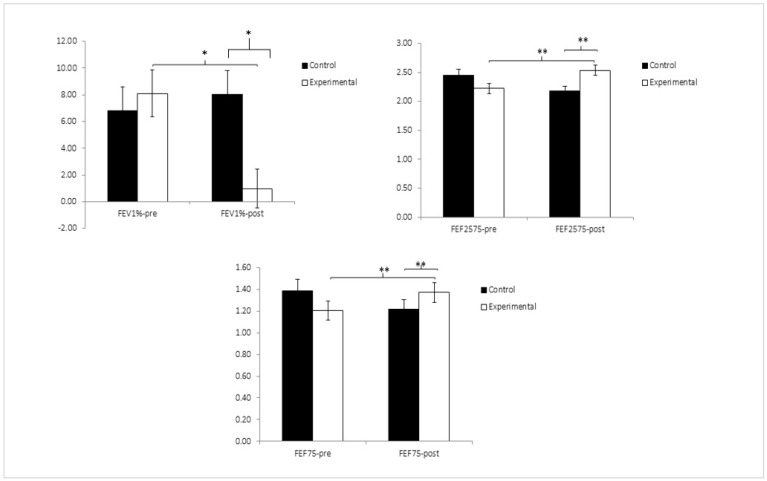
Inter- and intra-group comparisons regarding FEV1%, FEF25–75%, and FEF75%. FEV1%: ratio of FEV_1_/FVC; FEF25–75%: forced expiratory flow over middle half of forced vital capacity; FEF75%: forced expiratory flow at 75% of exhaled vital capacity. * *p* < 0.05, ** *p* < 0.01.

**Table 1 healthcare-12-00893-t001:** Technical foundations of hypopressive abdominal training.

Technical Foundations	Definition
Autoelongation	Axial stretching of the spine, tensioning of the deep spine and back extensors
Double chin	Pulling the crown to the ceiling
Decoaptation of the glenohumeral joint	Scapula abduction and serratus activation
Neutral pelvis	Equal distance between anterior and posterior superior iliac spines
Dorsal ankle flexion	Parallel lower extremities with hip width, slight knee flexion, and dorsal ankle flexion
Gravity shaft overrun	Imbalance of the anteroposterior axis involving variation in the center of gravity
Diaphragmatic breathing	Nasal inspiration focusing on the lateral expansion of the basal lung area, emphasizing enlargement of the lower rib spaces. Slow and controlled exhalation through the mouth
Expiratory apnea	Total exhalation Exhalation with open rib cage maintained while the diaphragm is returned to a position of relaxation through the in-drawing of the abdominal muscles, thereby lowering the intra-abdominal pressure (involuntary lifting of the pelvic floor)

**Table 2 healthcare-12-00893-t002:** Baseline characteristics of the study group.

Demographic Characteristics		All Participants (n = 117)	CG (n = 55)	EG (n = 62)	*p*-Value
Mean age ± SD		45.65 ± 8.86	46.89 ± 6.59	44.54 ± 10.40	0.149
Mean weight ± SD (kg)		63.59 ± 10.59	64.62 ± 10.04	62.67 ± 11.05	0.318
Mean height ± SD (cm)		162.56 ± 5.95	163.45 ± 5.83	161.78 ± 5.99	0.128
Mean BMI ± SD		24.03 ± 3.63	24.15 ± 3.31	23.93 ± 3.92	0.742
No. pregnant ± SD		1.54 ± 1.07	1.71 ± 1.06	1.38 ± 1.07	0.090
No. delivery ± SD		1.47 ± 1.05	1.63 ± 1.04	1.33 ± 1.05	0.130
Delivery type: n (%)	None	28 (23.5)	10 (35.7)	18 (64.3)	0.511
	Vaginal	68 (57.1)	34 (50)	34 (50)	0.824
	Caesarean	12 (10.1)	7 (58.3)	5 (41.7)	0.619
	Both	10 (8.4)	5 (45.5)	6 (54.5)	0.804
Smoking: n (%)	No	100 (84)	49 (49)	51 (51)	0.330
	Yes	19 (16)	7 (36.8)	12 (63.2)	0.563
Exercise: n (%)	No	56 (47.1)	27 (48.2)	29 (51.8)	0.812
	Yes	63 (52.9)	29 (46)	34 (54)	0.734

Notes: Values expressed as means and standard deviations and as frequencies and percentages for continuous and categorical variables. CG: control group; EG: experimental group; kg: kilograms; cm: centimeters; BMI: body mass index; No.: number.

**Table 3 healthcare-12-00893-t003:** Respiratory muscle function related to health.

	Experimental Group (n = 62)		Control Group (n = 55)		Group			Time			Group × Time		
	Pre Mean ± SD	Post Mean ± SD	Pre Mean ± SD	Post Mean ± SD	F(1,117)	*p*	η^2^	F(1,117)	*p*	η^2^	F(1,117)	*p*	η^2^
FVC	2.52 ± 0.41	2.91 ± 0.38	2.80 ± 0.41	2.75 ± 0.39	3.22	0.076	0.027	284.90	0.000	0.709	0.63	0.430	0.005
FEV_1_	1.98 ± 0.46	2.29 ± 0.45	2.16 ± 0.51	2.00 ± 0.44	0.37	0.546	0.003	3683.57	0.000	0.969	0.21	0.649	0.002
PEF	3.58 ± 1.15	4.02 ± 1.18	3.92 ± 1.42	3.55 ± 1.25	6.95	0.010	0.056	345.23	0.000	0.747	1.08	0.302	0.009
FEV1%	76.39 ± 14.01	79.11 ± 11.63	77.73 ± 13.41	73.48 ± 13.11	0.89	0.347	0.008	19.95	0.000	0.146	35.11	0.000	0.231
FEF25%	3.36 ± 1.07	3.72 ± 1.13	3.53 ± 1.35	3.18 ± 1.16	1.58	0.211	0.013	502.62	0.000	0.811	0.00	0.960	0.000
FEF75%	1.20 ± 0.45	1.37 ± 0.42	1.38 ± 0.48	1.22 ± 0.35	6.51	0.012	0.053	4696.97	0.000	0.976	5.73	0.018	0.047
FEF25–75%	2.22 ± 0.67	2.53 ± 0.71	2.44 ± 0.81	2.17 ± 0.65	7.54	0.007	0.061	601.86	0.000	0.837	5.62	0.019	0.046

Data are expressed as means ± standard deviations. FVC: forced vital capacity; FEV_1_: maximum volume expired in the first second of forced expiration; PEF: the peak expiratory flow rate; FEV1%: the ratio of FEV_1_/FVC; FEF25%: forced expiratory flow at 25% of expired vital capacity; FEF75%: forced expiratory flow at 75% of expired vital capacity; FEF25–75%: forced expiratory flow over the middle half of forced vital capacity.

**Table 4 healthcare-12-00893-t004:** Quality of life related to health.

SF-12	Experimental Group (n = 62)		Control Group (n = 55)		Group			Time			Group × Time		
	Pre Mean ± SD	Post Mean ± SD	Pre Mean ± SD	Post Mean ± SD	F(1,117)	*p*	η^2^	F(1,117)	*p*	η^2^	F(1,117)	*p*	η^2^
GH	66.67 ± 17.96	66.27 ± 16.29	62.95 ± 17.83	63.84 ± 17.14	1.18	0.279	0.010	0.03	0.865	<0.001	0.20	0.658	0.002
PF	89.29 ± 19.94	92.46 ± 16.58	87.50 ± 18.46	88.39 ± 19.05	1.01	0.317	0.009	1.33	0.251	0.011	0.42	0.519	0.004
PR	85.71 ± 30.36	91.27 ± 24.66	75.89 ± 39.30	86.61 ± 32.32	2.42	0.122	0.020	5.27	0.023	0.043	0.53	0.468	0.005
ER	67.46 ± 42.27	71.43 ± 43.73	78.57 ± 37.97	78.57 ± 40.29	1.84	0.178	0.015	0.33	0.568	0.003	0.33	0.568	0.003
BP	87.70 ± 18.98	92.06 ± 14.77	83.04 ± 25.27	91.52 ± 17.37	0.72	0.397	0.006	12.79	0.001	0.099	1.31	0.254	0.011
MH	66.03 ± 17.18	71.11 ± 17.14	69.11 ± 17.61	67.86 ± 14.98	0.00	0.972	<0.001	1.27	0.263	0.011	3.46	0.065	0.029
V	59.05 ± 21.98	67.30 ± 22.80	61.07 ± 23.95	68.57 ± 20.13	0.23	0.635	0.002	13.09	<0.001	0.101	0.03	0.863	<0.001
SF	83.33 ± 21.53	83.73 ± 25.46	83.48 ± 23.49	86.61 ± 17.81	0.20	0.653	0.002	0.56	0.456	0.005	0.34	0.336	0.089
MSC	84.06 ± 17.46	87.63 ± 13.24	78.79 ± 22.32	84.23 ± 19.00	2.30	0.132	0.019	6.88	0.010	0.056	0.29	0.294	0.084
PSC	68.23 ± 21.96	72.69 ± 22.62	73.32 ± 21.33	74.67 ± 19.13	1.01	0.317	0.009	2.85	0.094	0.024	0.81	0.812	0.145

Data are expressed as means ± standard deviations. GH: general health; PF: physical function; PR: physical role; ER: emotional role; BP: bodily pain; MH: mental health; V: vitality; SF: social function; MSC: mental summary component; PSC: physical summary component.

## Data Availability

Data are contained within the article.

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
