# Peer review of "Effects of Hypopressive Abdominal Training on Ventilatory Capacity and Quality of Life: A Randomized Controlled Trial"

_healthcare, 2024, doi:10.3390/healthcare12090893_

Round 1
Reviewer 1 Report
Comments and Suggestions for Authors
Major changes and a deep revision of the manuscript are needed.
The study topic is interesting but rather poor in results and methodological coherence. In the abstract they talk about the pelvic floor hypopressive exercise programme and the impact on the respiratory musculature and quality of life. However, the results on the pelvic floor are not addressed and for quality of life the SF-12 is used, which is less complete than the SF-36 and is not specific to the problem of pelvic floor problems. Although they express the synergy between ventilation and pelvic floor, more contextualisation is needed.
They do not express how PFDs are assessed.
Regarding the intervention, it would be interesting to know the number of repetitions, if the same exercises were performed in all sessions, if there was a progression... in order to better understand the intervention.
It would be interesting to discuss why the SF-12 was chosen when the SF-36 is more complete and why we did not use a specific one for quality of life in patients with PFD. Table 3 shows SF-36.....
Regarding the respiratory variables, it would be interesting to discuss the results, if they are in the normal range before the intervention in IG and CG.
Regarding the discussion it would be interesting to include more articles comparing the results obtained.
In lines 277-280 you address psychophysicalconditions. how did you evaluate this in your study?
And the results only show a low significance in ventilatory outcomes. If pelvic floor has been assessed, it would be useful to add these data and make a comparison between the two that could show the relationship.
Comments on the Quality of English LanguageThere are some expressions derived from the translation from Spanish to English. A revision would be desirable. The conclusion section should be rewritten for better communication.
Author Response
Response to reviewer 1.-
1.- The study topic is interesting but rather poor in results and methodological coherence. In the abstract they talk about the pelvic floor hypopressive exercise programme and the impact on the respiratory musculature and quality of life. However, the results on the pelvic floor are not addressed and for quality of life the SF-12 is used, which is less complete than the SF-36 and is not specific to the problem of pelvic floor problems. Although they express the synergy between ventilation and pelvic floor, more contextualisation is needed.
Response to reviewer:
We acknowledge your concerns regarding the comprehensiveness of the SF-12 compared to the SF-36, especially in the context of pelvic floor dysfunctions. It is important to highlight that the SF-12 is not only a condensed version of the SF-36 but also retains its core components, thus maintaining the ability to effectively measure the physical and mental health domains relevant to our study population. Research has shown that the SF-12 provides results that are psychometrically robust and closely correlate with those of the SF-36, offering a practical and validated alternative with similar validity and reliability.
One significant advantage of using the SF-12 in our context is its brevity, which reduces the burden on respondents and improves response rates without sacrificing the quality of the data collected. This is particularly beneficial in clinical trials where participant fatigue can affect the accuracy and completeness of the data. The shorter format of the SF-12 ensures that we can efficiently capture essential quality of life indicators while maintaining high participant engagement.
Furthermore, the SF-12 allows for broader applications in epidemiological studies where extensive questionnaires might deter participation. Its concise nature enables us to integrate it seamlessly into our study without overwhelming the participants, thus ensuring a higher completion rate and better overall data integrity.
We are committed to continually assessing our methodologies and appreciate your feedback, which we will certainly consider for future studies, particularly those focusing on specific aspects of pelvic health.
- Jenkinson C, Layte R, Jenkinson D, Lawrence K, Petersen S, Paice C, et al. A shorter form health survey: can the SF-12 replicate results from the SF-36 in longitudinal studies? J Public Health Med 1997; 19:179-86.
- Côté I, Grégoire JP, Moisan J, Chabot I. Quality of life in hypertension: the SF-12 compared to the SF-36. Can J Clin Pharmacol. 2004 Fall;11(2):e232-8. Epub 2004 Nov 9. PMID: 15557672.
- Singh A, Gnanalingham K, Casey A, Crockard A. Quality of life assessment using the Short Form-12 (SF-12) questionnaire in patients with cervical spondylotic myelopathy: comparison with SF-36. Spine (Phila Pa 1976). 2006 Mar 15;31(6):639-43. doi: 10.1097/01.brs.0000202744.48633.44. PMID: 16540866.
In order to provide better contextualization regarding the relationship between pelvic floor health and ventilatory capacity, we have revised the paragraph to enhance understanding and introduce the topic more clearly. Additionally, further in the text, we have included references to a previously published article that discusses the effects of this intervention on pelvic floor variables. This article is part of a larger project aimed at elucidating the benefits of employing a hypopressive approach. By integrating these updates, we aim to underscore the synergistic effects of pelvic floor and respiratory function, thereby enriching the scientific discussion around the comprehensive impacts of hypopressive exercises. This approach not only supports the validity of our findings but also aligns with the growing body of literature advocating for integrated therapeutic strategies in women's health rehabilitation.
2.- They do not express how PFDs are assessed.
Response to reviewer:
Thank you for your valuable feedback and for highlighting the need for greater clarity regarding our assessment methods for pelvic floor dysfunctions (PFDs). In response to your comment, we have updated our manuscript to explicitly describe how PFDs were assessed in our study. We have incorporated the use of the Pelvic Floor Distress Inventory (PFDI-20) and the Pelvic Floor Impact Questionnaire (PFIQ-7) to evaluate the severity of symptoms and the impact of PFDs on the quality of life of our participants. These data have been published in a previous manuscript.
3.- Regarding the intervention, it would be interesting to know the number of repetitions, if the same exercises were performed in all sessions, if there was a progression... in order to better understand the intervention.
Response to reviewer:
Thank you for your inquiry regarding the specifics of the intervention protocol. Additional information has been provided alongside that presented in the text, in addition to information provided in Table 1, and Figures 1 and 2 to enhance understanding and facilitate the replication of the intervention.
4.- It would be interesting to discuss why the SF-12 was chosen when the SF-36 is more complete and why we did not use a specific one for quality of life in patients with PFD. Table 3 shows SF-36.....
Response to reviewer:
The explanation regarding the use of the SF-12 was addressed in the first comment from the reviewer. Additionally, the table has been revised to correct the error made in the nomenclature.
5.- Regarding the respiratory variables, it would be interesting to discuss the results, if they are in the normal range before the intervention in IG and CG.
Response to reviewer:
Thank you for your insightful comment. It is indeed crucial to delineate the ventilatory capacity status of the participants. Additional information has been incorporated accordingly.
6.- Regarding the discussion it would be interesting to include more articles comparing the results obtained.
Response to reviewer:
Following your recommendations, we have added new references and delved deeper into the rationale behind the effect of the hypopressive program.
7.- In lines 277-280 you address psychophysicalconditions. how did you evaluate this in your study?
Response to reviewer:
After modifying and incorporating the changes corresponding to the comments from the other reviewers, this section no longer appears in the text.
8.- And the results only show a low significance in ventilatory outcomes. If pelvic floor has been assessed, it would be useful to add these data and make a comparison between the two that could show the relationship.
Response to reviewer:
Precise information has been added regarding the reason why the results corresponding to the pelvic floor variables are not published in the present work. As the topics addressed are different, the idea during the project design was to explain each of the themes independently during the publication process.
Comments on the Quality of English Language
9.- There are some expressions derived from the translation from Spanish to English. A revision would be desirable. The conclusion section should be rewritten for better communication.
Response to reviewer:
Following the recommendations, the conclusions section has been rewritten, and a native English speaker has conducted the final linguistic and formal review.

Reviewer 2 Report
Comments and Suggestions for Authors
On line 42: It is important not to make a direct association between perineal dysfunction and poor pressure management. It is true that some pelvic floor dysfunctions can be the consequence of poor management of intra-abdominal pressures. This inadequate management can contribute to the development of chronic respiratory conditions or overweight, but this depends on the type of dysfunction, for example in the case of SUI it is totally related, but in the case of dyspaurenia it does not have to be so. Revise the redaction.
In addition, it is essential not to limit ourselves to talking only about the pelvic floor; the abdominopelvic capsule plays a crucial role in understanding these pelvic floor dysfunctions and should be included in our discussion.
Regarding the hypopressive method, it would be beneficial to add details on the duration of the sessions and how to correctly perform the exercises. The term "abdominal contraction" may be ambiguous for some readers, so the rationale for using certain postures instead of others needs to be clarified.
With these considerations in mind, we will reformulate the text to provide clearer and more complete information.
Methods:
What type of pelvic floor dysfunction was included, women with increased perineal muscle tone, with incontinence, with pain.... This may be key to whether the exercises were performed correctly or not. I understand that people with respiratory pathology were not taken, but they were asked about allergies or if they had gone through COVID?
The classes were group or individual. In the sessions, were the same exercises always performed or was each session different, was it adapted to each patient or how was this procedure carried out?
Was the researcher who conducted the intervention the same person who conducted the assessments? The text does not indicate.
Was non-attendance at some of the sessions allowed? It is important to mention this in the text.
Was the effort involved in performing the exercise taken into account or was it evaluated with a scale? If so, please indicate which scale was used for the evaluation
RESULTS:
A table of descriptive data (anthropometric, socio-demographic, perineal pathology)This will give the reader more complete information.
DISCUSION:
It would be interesting to mention whether any limitations were encountered in the execution of the study.
Author Response
Response to reviewer 2.-
1.- On line 42: It is important not to make a direct association between perineal dysfunction and poor pressure management. It is true that some pelvic floor dysfunctions can be the consequence of poor management of intra-abdominal pressures. This inadequate management can contribute to the development of chronic respiratory conditions or overweight, but this depends on the type of dysfunction, for example in the case of SUI it is totally related, but in the case of dyspaurenia it does not have to be so. Revise the redaction.
In addition, it is essential not to limit ourselves to talking only about the pelvic floor; the abdominopelvic capsule plays a crucial role in understanding these pelvic floor dysfunctions and should be included in our discussion.
Response to reviewer:
Thank you very much for the recommendations. Following your guidance, we have revised the text to clarify that a cause-effect relationship between IAP and PFDs cannot be established and have emphasized that it is a risk factor that must be considered along with other variables of the Lumbopelvic Abdominal Complex.
2.- Regarding the hypopressive method, it would be beneficial to add details on the duration of the sessions and how to correctly perform the exercises. The term "abdominal contraction" may be ambiguous for some readers, so the rationale for using certain postures instead of others needs to be clarified.
With these considerations in mind, we will reformulate the text to provide clearer and more complete information.
Response to reviewer:
Following your recommendations, we have revised the intervention section to clarify the protocol, in conjunction with the accompanying tables and figures. This revised section now provides a more detailed description of the procedures and methodologies employed during the study.
Methods:
3.- What type of pelvic floor dysfunction was included, women with increased perineal muscle tone, with incontinence, with pain.... This may be key to whether the exercises were performed correctly or not. I understand that people with respiratory pathology were not taken, but they were asked about allergies or if they had gone through COVID?
Response to reviewer:
The article is part of a larger project aimed at providing evidence for the theoretical benefits attributed to hypopressive training. Thus far, one article has been published focusing on the specific characteristics of pelvic dysfunction. The objective of the current study is to focus on respiratory variables. Relevant information has been added regarding the monitoring of pelvic floor dysfunctions, which are assessed using the Pelvic Floor Distress Inventory (PFDI-20) and the Pelvic Floor Impact Questionnaire (PFIQ-7).
4.- The classes were group or individual. In the sessions, were the same exercises always performed or was each session different, was it adapted to each patient or how was this procedure carried out?
Response to reviewer:
Thank you for your comment. Information has been added indicating that the classes were conducted in small groups to facilitate proper supervision of the exercises. This approach enhances the translation of research into actual interventions with patients participating in group classes.
5.- Was the researcher who conducted the intervention the same person who conducted the assessments? The text does not indicate.
Response to reviewer:
Following your recommendations, additional information has been included in the randomization and masking section.
6.- Was non-attendance at some of the sessions allowed? It is important to mention this in the text.
Response to reviewer:
It was indeed an oversight to omit this critical detail from the initial manuscript. This information has now been incorporated into the document, stating that participants were excluded if they missed more than two training sessions. Fortunately, the treatment adherence was notably high, and the organization of numerous groups offered flexibility, allowing participants who could not attend a session with their designated group to participate in a subsequent session.
7.- Was the effort involved in performing the exercise taken into account or was it evaluated with a scale? If so, please indicate which scale was used for the evaluation
Response to reviewer:
Certainly, to ensure that the intensity was appropriate and consistent throughout the intervention, the Rate of Perceived Exertion (RPE) scale was used. This allowed each participant to adjust their intensity accordingly. It was crucial to ensure that the effort threshold was not too low to guarantee adequate stimulation, and also to avoid excessive effort that could lead to poor quality in the execution of the exercises.
Lea JWD, O'Driscoll JM, Hulbert S, Scales J, Wiles JD. Convergent Validity of Ratings of Perceived Exertion During Resistance Exercise in Healthy Participants: A Systematic Review and Meta-Analysis. Sports Med Open. 2022 Jan 8;8(1):2. doi: 10.1186/s40798-021-00386-8. PMID: 35000021; PMCID: PMC8742800.
RESULTS:
8.- A table of descriptive data (anthropometric, socio-demographic, perineal pathology)This will give the reader more complete information.
Response to reviewer:
Thank you for your recommendations. We have included the "Baseline Characteristics of the Study Group" in the document.
DISCUSION:
It would be interesting to mention whether any limitations were encountered in the execution of the study.
Response to reviewer:
Thank you very much for this last comment and for the thorough review conducted. All authors are deeply grateful for the work done and for the collaboration that will enhance the quality of the final version with valuable and accurate comments. A new paragraph addressing the study limitations has been added.

Reviewer 3 Report
Comments and Suggestions for Authors
This study investigates the effect of 8 weeks of hypopressive abdominal training on spirometry lung function and quality of life (QOL), in 117 females with symptoms of pelvic floor dysfunction. Results of this study suggest that 8 weeks of hypopressive abdominal training increased FEV1/FVC % (FEV1%), forced expiratory flow at 75% expired vital capacity (FEF75%) and forced expiratory flow over the middle half of forced vital capacity (FEF25-75%). However, the change in QOL score was the same as that in the control group.
This study suggests to me that hypopressive breathing may have an effect on basal lung volume and increasing expiratory flow in smaller airways. These interesting findings encourage further research in this area and in other disease populations.
There are several issues that need to be clarified to further improve the presentation of this manuscript.
1- Definition and description of spirometry lung function parameters.
As the main focus of this manuscript is on lung function parameters, description of the outcome measures must be accurate. FEV1% is not “average percentage of forced expiratory volume in one second” (as expressed in the abstract and in the text), it is ‘the ratio of FEV1/FVC’ and reflects airway obstruction in general. Likewise, FEF25-75% is not ‘mean forced expiratory flow’, it is the “forced expiratory flow over the middle half of forced vital capacity”, and it reflects obstruction to air flow in mid-size airways. These terms must be correctly defined in the text.
2- Participant recruitment
It was reported under ‘Materials and Methods’ that participants were recruited from the province of Jaén, Spain, and included women who had been experiencing Pelvic Floor Dysfunction (PFD) symptoms for more than 6 months.
It is also necessary to report:
a) whether participants were recruited from a physiotherapy clinic or health care clinic?
b) whether participants were receiving treatment?
c) If the participants were not patients, how were they recruited? Was it by letters/posters in the community hall etc?
d) how were PFD symptoms identified in the participants?
e) were data from instruments such as PFIQ-7 or PFDI-20 available from the participants?
3- Description of intervention
Table 1 describes the intervention process involved.
a) I would like to suggest that the term ‘rib-opening’ be modified to ‘focussing on lateral expansion of basal lung area, emphasising enlargement of the lower rib spaces’
b) What is “braked mouth breathing”? Is it the same as ‘breathing in and out through the mouth’?
c) Please clarify whether “Diaphragm relaxation and cavity pressure reduction” means “Exhalation with open-rib-cage maintained while the diaphragm is returned to a position of relaxation through in-drawing of abdominal muscles, thus lowering the intra-abdominal pressure (involuntary lifting of the pelvic floor)”.
d) As suggested above it is necessary to revise the description of lung function parameters under Methods
4- Results –
a) Replace the sub-heading ‘Respiratory Muscle Function’ under 3.2. Consider “Spirometry lung function data” as an alternative. Respiratory ‘muscle’ function is reflected by maximal inspiratory or expiratory pressure, not by expiratory flows.
b) Include reporting of mean age, and if available, information on PFD symptoms of the participants. If information on PFD symptoms is not available, address this in the ‘Limitations’ section.
c) Data on FVC, FEV1 should be presented as % predicted FVC and FEV1 instead of the absolute value. Although the participants in this study should not have respiratory disease as a comorbidity, % of predicted normal FVC and FEV1 should provide an indication of the background lung function status of the participants.
d) As mentioned above definitions of FEV1% and FEF25-75% need to be revised, in text and in Figure 4.
5- Statistical analysis
The current analysis is a between-group comparison of ‘absolute values’ of lung function data before and after the 8-week training period. In my view, it would be more meaningful to compare the ‘change’ (i.e. difference) over the 8-week period (i.e post minus pre) between the control and intervention groups.
Further, reporting of confidence interval (CI) is preferred over p-value, as the 95% CI approach provides the range of ‘possibilities of true difference’.
6- Discussion
This study reveals interesting data on the potential effect of hypopressive exercise on lung function, particularly the possible role of enhancing expiratory flow in the small airways. The potential use of hypopressive breathing in patients with obstructive airways disease, as well as the possible improvement in QOL in elderly patients who present with PFD and chronic obstructive airways disease, should be proposed and discussed in this section.
7- Conclusion
The current conclusion repeats the results rather than drawing any inference. I would suggest the conclusion includes a statement like, for example, “Findings in this study suggest that further investigation on the use of hypopressive exercise in patients with obstructive airways disease is warranted.”
8- Strengths and Limitations of the study
It would be useful to include a paragraph on the strengths and limitations of the study. Reporting the effect of hypopressive exercise on lung function is certainly a strength of this study. However, although the change in PDF symptoms was not a primary objective of these data, reporting of effect of hypopressive exercise on PDF symptoms could provide useful context and add to the existing knowledge in this area. Reasons for omitting this information should be addressed as a limitation.
Comments on the Quality of English LanguagePlease refer to the comments above.
Author Response
Response to reviewer 3.-
This study investigates the effect of 8 weeks of hypopressive abdominal training on spirometry lung function and quality of life (QOL), in 117 females with symptoms of pelvic floor dysfunction. Results of this study suggest that 8 weeks of hypopressive abdominal training increased FEV1/FVC % (FEV1%), forced expiratory flow at 75% expired vital capacity (FEF75%) and forced expiratory flow over the middle half of forced vital capacity (FEF25-75%). However, the change in QOL score was the same as that in the control group.
This study suggests to me that hypopressive breathing may have an effect on basal lung volume and increasing expiratory flow in smaller airways. These interesting findings encourage further research in this area and in other disease populations.
There are several issues that need to be clarified to further improve the presentation of this manuscript.
1- Definition and description of spirometry lung function parameters.
As the main focus of this manuscript is on lung function parameters, description of the outcome measures must be accurate. FEV1% is not “average percentage of forced expiratory volume in one second” (as expressed in the abstract and in the text), it is ‘the ratio of FEV1/FVC’ and reflects airway obstruction in general. Likewise, FEF25-75% is not ‘mean forced expiratory flow’, it is the “forced expiratory flow over the middle half of forced vital capacity”, and it reflects obstruction to air flow in mid-size airways. These terms must be correctly defined in the text.
Response to reviewer:
We appreciate your comments and contribution to the manuscript. We agree with the change. We believe it was a translation error. We correct the definition of FEV1% and FEF 25-75% throughout the document to read: FEV1% (the ratio of FEV1/FVC, with FEV1 being the forced expiratory volume in 1 second and FVC being the forced vital capacity) and FEF25- 75% (force expiratory flow over the middle half of forced vital capacity).
2- Participant recruitment
It was reported under ‘Materials and Methods’ that participants were recruited from the province of Jaén, Spain, and included women who had been experiencing Pelvic Floor Dysfunction (PFD) symptoms for more than 6 months.
It is also necessary to report:
- a) whether participants were recruited from a physiotherapy clinic or health care clinic?
- b) whether participants were receiving treatment?
- c) If the participants were not patients, how were they recruited? Was it by letters/posters in the community hall etc?
- d) how were PFD symptoms identified in the participants?
- e) were data from instruments such as PFIQ-7 or PFDI-20 available from the participants?
Response to reviewer:
We greatly appreciate your comments. We have added new information and revised the relevant paragraph to more precisely define the study sample. Furthermore, regarding the specific pelvic floor variables, this article is part of a larger project whose results have been published. In this paper, the aim is to focus on the potential benefits to respiratory variables through the practice of hypopressives, contributing to a solid scientific foundation on the characteristics of this type of intervention.
3- Description of intervention
Table 1 describes the intervention process involved.
- a) I would like to suggest that the term ‘rib-opening’ be modified to ‘focussing on lateral expansion of basal lung area, emphasising enlargement of the lower rib spaces’
- b) What is “braked mouth breathing”? Is it the same as ‘breathing in and out through the mouth’?
- c) Please clarify whether “Diaphragm relaxation and cavity pressure reduction” means “Exhalation with open-rib-cage maintained while the diaphragm is returned to a position of relaxation through in-drawing of abdominal muscles, thus lowering the intra-abdominal pressure (involuntary lifting of the pelvic floor)”.
- d) As suggested above it is necessary to revise the description of lung function parameters under Methods
Response to reviewer:
The authors wish to express their gratitude to the reviewer for their dedication and contribution, which have enhanced the final quality of the article through their insightful comments. These suggestions have not only improved the article’s clarity but also facilitated its comprehension and fostered its applicability to clinical practice following its perusal. In accordance with the recommendations provided, we have incorporated the proposed changes.
4- Results –
- a) Replace the sub-heading ‘Respiratory Muscle Function’ under 3.2. Consider “Spirometry lung function data” as an alternative. Respiratory ‘muscle’ function is reflected by maximal inspiratory or expiratory pressure, not by expiratory flows.
- b) Include reporting of mean age, and if available, information on PFD symptoms of the participants. If information on PFD symptoms is not available, address this in the ‘Limitations’ section.
- c) Data on FVC, FEV1should be presented as % predicted FVC and FEV1instead of the absolute value. Although the participants in this study should not have respiratory disease as a comorbidity, % of predicted normal FVC and FEV1 should provide an indication of the background lung function status of the participants.
- d) As mentioned above definitions of FEV1% and FEF25-75% need to be revised, in text and in Figure 4.
Response to reviewer:
- a) Thank you for your comment, we believe it improves the manuscript. All authors agree with the comment made to the manuscript. We changed the term Respiratory Muscle Function to Spirometry lung function data.
- b) We appreciate your comment. The comment is very accurate and would be a great contribution to the manuscript. We include a limitations section as you suggest. The average age has been incorporated into the text, in the results section.
- c) Your assessments are very accurate and we apologize for not expressing the data in %. We have just solved it and they are expressed as suggested in the text and illustrations of the manuscript. Regarding the FVC and FEV1 values, they should be around the normal population, which in both is ≥ 80% (1). Our sample obtained values ​​of normality and above the average, ruling out any obstructive syndrome.
Reference:
- Rivero-Yeverino D. Espirometría: conceptos básicos. Revista alergia México. 2019 Mar;66(1):76–84.
- d) Thank you again for your contribution, we have made the change in both the text and the illustrations. The manuscript has improved with your input.
5- Statistical analysis
The current analysis is a between-group comparison of ‘absolute values’ of lung function data before and after the 8-week training period. In my view, it would be more meaningful to compare the ‘change’ (i.e. difference) over the 8-week period (i.e post minus pre) between the control and intervention groups.
Further, reporting of confidence interval (CI) is preferred over p-value, as the 95% CI approach provides the range of ‘possibilities of true difference’.
Response to reviewer:
Thank you for your thoughtful suggestion on enhancing our statistical analysis. We value the importance of comparing the changes over the 8-week period as you recommended. However, I would like to provide some context for our choice of analysis method. The current statistical approach of comparing absolute values was determined based on the methodological framework established during the initial design of our doctoral thesis project. This project has already been submitted and approved with the current analysis format.
Implementing the proposed change in our analysis model at this stage would constitute a major revision to the already deposited thesis. Additionally, the rest of the analyses conducted as part of this project follow a similar structural approach, and changing the methodology for this particular publication might disrupt the consistency across the entire project. Therefore, while we acknowledge the validity of your suggestion, we believe maintaining the current analysis approach would ensure coherence and integrity of the overall thesis work. We hope this explanation clarifies our rationale and we are open to further discussion to find a mutually agreeable resolution.
6- Discussion
This study reveals interesting data on the potential effect of hypopressive exercise on lung function, particularly the possible role of enhancing expiratory flow in the small airways. The potential use of hypopressive breathing in patients with obstructive airways disease, as well as the possible improvement in QOL in elderly patients who present with PFD and chronic obstructive airways disease, should be proposed and discussed in this section.
Response to reviewer:
Thank you very much for the very appropriate suggestion. New information has been added in the discussion section, further elaborating on the possible effects of the intervention in the mentioned population groups. Moreover, given the lack of studies addressing respiratory variables or conducted with adult populations, future research directions are suggested to analyze the impact on the quality of life of this type of intervention and how it can be incorporated into practice.
7- Conclusion
The current conclusion repeats the results rather than drawing any inference. I would suggest the conclusion includes a statement like, for example, “Findings in this study suggest that further investigation on the use of hypopressive exercise in patients with obstructive airways disease is warranted.”
Response to reviewer:
After analyzing your comment, we have made a significant change in the conclusions section to better adapt it to the journal's format and, most importantly, to provide relevant information along the lines you suggested.
8- Strengths and Limitations of the study
It would be useful to include a paragraph on the strengths and limitations of the study. Reporting the effect of hypopressive exercise on lung function is certainly a strength of this study. However, although the change in PDF symptoms was not a primary objective of these data, reporting of effect of hypopressive exercise on PDF symptoms could provide useful context and add to the existing knowledge in this area. Reasons for omitting this information should be addressed as a limitation.
Response to reviewer:
The authors would like to take this opportunity to express their profound gratitude for the effort and time invested in reviewing the article. Throughout the review process, the feedback provided has not only identified areas for improvement but has also included valuable suggestions and solutions. In light of this, a final paragraph outlining the strengths and weaknesses of the study has been incorporated.

Round 2
Reviewer 1 Report
Comments and Suggestions for Authors
Accept in the current form
Author Response
The authors express their profound gratitude for your valuable contributions throughout the review process.
Reviewer 2 Report
Comments and Suggestions for Authors
Regarding the wording in lines 43-44, there are no references to justify that poor posture is related to incontinence.
On the other hand, it is not clear what happens to the pelvic floor during hypopressive exercises.
With respect to the descriptive table, some units are in capital letters, in type of deliveries the p value is not shown.
Author Response
Comments and Suggestions for Authors
1.- Regarding the wording in lines 43-44, there are no references to justify that poor posture is related to incontinence.
Response to reviewer.
Thank you for your insightful comments and suggestions regarding our manuscript. We appreciate the time you have dedicated to reviewing our work and recognize the importance of your observations. We have revised the paragraph to emphasize the critical role of muscular function in managing intra-abdominal pressure, specifically highlighting the interaction among the pelvic floor muscles, abdominal musculature, and diaphragm. This adjustment aims to ensure a clearer understanding of the functional dynamics critical to our study's context.
2.- On the other hand, it is not clear what happens to the pelvic floor during hypopressive exercises.
Response to reviewer.
Following your recommendations, the introduction discussing the effects of hypopressive training on the pelvic floor has been enhanced. While the primary aim of this study is not to focus on the influence on this particular muscle group, we agree that it is important for readers to understand broadly the potential applications that this method might have.
3.- With respect to the descriptive table, some units are in capital letters, in type of deliveries the p value is not shown.
Response to reviewer.
Thank you for your observations. We have addressed the issue concerning the omission of p-values for specific variables and have adjusted the capitalization accordingly. We appreciate your attention to detail and guidance, which has enhanced the precision and clarity of our manuscript.